# Antibody neutralization of SARS-CoV-2 through ACE2 receptor mimicry

Jiwan Ge [1,5], Ruoke Wang [2,5], Bin Ju[3,5], Qi Zhang[2,5], Jing Sun[4], Peng Chen [2], Senyan Zhang [1], Yuling Tian [1], Sisi Shan [2], Lin Cheng[3], Bing Zhou[3], Shuo Song[3], Juanjuan Zhao[3], Haiyan Wang[3], Xuanling Shi[2], Qiang Ding [2], Lei Liu[3], Jincun Zhao [4], Zheng Zhang [3✉], Xinquan Wang[1✉] & Linqi Zhang [2✉]

Understanding the mechanism for antibody neutralization of SARS-CoV-2 is critical for the development of effective therapeutics and vaccines. We recently isolated a large number of monoclonal antibodies from SARS-CoV-2 infected individuals. Here we select the top three most potent yet variable neutralizing antibodies for in-depth structural and functional analyses. Crystal structural comparisons reveal differences in the angles of approach to the receptor binding domain (RBD), the size of the buried surface areas, and the key binding residues on the RBD of the viral spike glycoprotein. One antibody, P2C-1F11, most closely mimics binding of receptor ACE2, displays the most potent neutralizing activity in vitro and conferred strong protection against SARS-CoV-2 infection in Ad5-hACE2-sensitized mice. It also occupies the largest binding surface and demonstrates the highest binding affinity to RBD. More interestingly, P2C-1F11 triggers rapid and extensive shedding of S1 from the cell-surface expressed spike glycoprotein, with only minimal such effect by the remaining two antibodies. These results offer a structural and functional basis for potent neutralization via disruption of the very first and critical steps for SARS-CoV-2 cell entry.

[1] The Ministry of Education Key Laboratory of Protein Science, Beijing Advanced Innovation Center for Structural Biology, Beijing Frontier Research Center for Biological Structure, Collaborative Innovation Center for Biotherapy, School of Life Sciences, Tsinghua University, Beijing 100084, China. [2] Comprehensive AIDS Research Center, Beijing Advanced Innovation Center for Structural Biology, School of Medicine and Vanke School of Public Health, Tsinghua University, Beijing 100084, China. [3] Institute for Hepatology, National Clinical Research Center for Infectious Disease, Shenzhen Third People's Hospital; The Second Affiliated Hospital, School of Medicine, Southern University of Science and Technology, Shenzhen 518112 Guangdong Province, China. [4] State Key Laboratory of Respiratory Disease, National Clinical Research Center for Respiratory Disease, Guangzhou Institute of Respiratory Health, the First Affiliated Hospital of Guangzhou Medical University, Guangzhou, Guangdong 510182, China. [5] These authors contributed equally: Jiwan Ge, Ruoke Wang, Bin Ju, Qi Zhang. ✉email: zhangzheng1975@aliyun.com; xinquanwang@mail.tsinghua.edu.cn; zhanglinqi@tsinghua.edu.cn

Novel coronavirus disease (COVID-19) is caused by SARS-CoV-2, a new member of the human betacoronavirus family that includes severe acute respiratory syndrome coronavirus (SARS-CoV) and middle east respiratory syndrome coronavirus (MERS-CoV)[1–3]. The virus was initially identified in Wuhan, China in early 2020 and has become a global pandemic with no available treatments or vaccines. Like other coronaviruses, the spike (S) glycoprotein of SARS-CoV-2 mediates viral entry and also serves as a target for neutralizing antibodies (nAbs). This type I fusion protein has a prefusion metastable homotrimer structure. Each monomer consists of noncovalently bound S1 and S2 subunits. Upon binding to the receptor angiotensin-converting enzyme 2 (ACE2), the S1 subunit undergoes a natural shedding process and exposes the S2 subunit to form a stabilized postfusion conformation[4,5]. Interestingly, the receptor-binding domain (RBD) of S1 experiences spontaneous "up" and "down" conformations where only the "up" position is accessible by receptor ACE2[4–10]. However, the "up" conformation is believed to be less stable, which may explain why the dominant trimer state has only one of the three RBDs standing up[9,10].

Researchers have discovered a growing number of anti-SARS-CoV-2 nAbs, providing candidates for therapeutics and guidance for vaccine design[11–22]. Broadly speaking, these nAbs recognize RBD, N-terminal domain (NTD), and other regions on the S glycoprotein that directly or indirectly interfere with the ACE2 interaction. The majority are SARS-CoV-2 specific, although a few cross-neutralize both SARS-CoV-2 and SARS-CoV[18,23–25]. The specific neutralizers identified to date derive exclusively from SARS-CoV-2 infected individuals, whereas the cross neutralizers are mostly from SARS-CoV infected or immunized animals[18,23–25]. This suggests RBDs from SARS-CoV-2 and SARS-CoV are immunologically distinct despite having near-identical conformational structures[26–29]. Therapeutic strategies must therefore target each species differently to achieve maximum efficacy.

Recently, we reported the isolation and characterization of a large number of RBD-specific monoclonal antibodies (mAbs) derived from single B cells of eight SARS-CoV-2 infected individuals. Among those, P2C-1F11, P2C-1A3, and P2B-2F6 demonstrated the most potent yet variable neutralizing activity against live SARS-CoV-2 without cross-reactivity with SARS-CoV[14]. However, the structural and functional basis for such differences remained unknown. P2C-1F11 was the strongest neutralizer ($IC_{50} = 0.03\ \mu g\ mL^{-1}$), followed by P2C-1A3 ($IC_{50} = 0.28\ \mu g\ mL^{-1}$), and P2B-2F6 ($IC_{50} = 0.41\ \mu g\ mL^{-1}$). Crystal structure analysis of the RBD–P2B–2F6 complex revealed moderate steric hindrance expected to interfere with viral engagement with ACE2, thereby interrupting viral entry[14]. Here, we determine the crystal structures of P2C-1F11 and P2C-1A3 bound to the SARS-CoV-2 RBD at a resolution of 2.95 Å and 3.40 Å, respectively (Supplementary Table 1). We also run head-to-head comparisons of all three structures to better understand the structural and functional basis for their differences in neutralizing activities.

## Results

### P2C-1F11 resembles ACE2 in binding to SARS-CoV-2 RBD.
The three antibodies approach the RBD from different angles and are therefore expected to have varying degrees of clashes with ACE2 when bound to the RBD (Fig. 1a). P2C-1F11 bound to RBD most resembles RBD-ACE2 binding, with only 25° of inter-angle deviation to the right of RBD-ACE2 (Fig. 1b). Consequently, the total volume of clashed residues between P2C-1F11 and ACE2 when both would bind to the same RBD molecule (clash volume) reached to ~20,480 Å³, which is the largest among

the three antibodies. Furthermore, the total number of residues shared between the P2C-1F11 epitope and ACE2-binding site is as high as 11 on the RBD (K417, Y453, L455, F456, A475, F486, N487, Y489, Q493, G502, and Y505) (Fig. 1c). In comparison, P2C-1A3 deviates to the left of ACE2 with an increased inter-angle of 42° (Fig. 1b). As a result, the estimated clash volume between the two decreases significantly to ~6860 Å³, and the number of shared RBD-binding residues decreases to nine (G446, Y449, F456, F486, N487, Y489, Q493, Q498, and T500) (Fig. 1c). In addition, P2B-2F6 deviates further to the left of ACE2 with an inter-angle increasing to 51°, more than double that of P2C-1F11 (Fig. 1b). The calculated clash volume, therefore, decreases to ~330 Å³, the smallest among the three antibodies. Only two residues, G446 and Y449, are shared between the ACE2-binding site and P2B-2F6 epitope on the RBD (Fig. 1c). It is notable that P2C-1F11 is the closest in resembling of ACE2 binding and the strongest in neutralizing activity among the three nAbs studied here. This provides some evidence that P2C-1F11 exerts its neutralizing activity through receptor functional mimicry.

### P2C-1F11 has the largest binding interface on SARS-CoV-2 RBD.
Binding interfaces varied significantly among the three antibodies. A total of 22 residues in the P2C-1F11 paratope interact with the RBD, 16 of which are derived from the heavy chain (6 from HCDR1; 5 from HCDR2; 5 from HCDR3) and only 6 from the light chain (4 from LCDR1; 2 from LCDR3) (Fig. 2a and Supplementary Table 2). As a result, P2C-1F11 buries the largest surface area of 955 Å² on the RBD among the three antibodies, 725 Å² of which was derived from the heavy chain and 230 Å² from the light chain. The largest binding surface translated into the tightest binding of P2C-1F11 to the RBD ($K_D = 1.72 \pm 0.84$ nM) among the three antibodies (Supplementary Fig. 1), which is largely mediated by a mixture of hydrophobic and hydrophilic interactions through HCDR1–3 and LCDR1 (Fig. 2b). Similarly, P2C-1A3 prefers heavy chain-mediated binding. Of the 21 paratope residues that interact with the RBD, 14 are from the heavy chain (1 from HCDR1; 5 from HCDR2; 4 from HFR3; 4 from HCDR3) and 7 from the light chain (3 from LCDR1; 4 from LCDR3) (Fig. 2c and Supplementary Table 2). The buried RBD surface area is 891 Å², of which 590 Å² is from the heavy chain and 301 Å² from light chain regions. The relatively smaller interface may account for the weaker binding affinity compared to P2C-1F11 as measured by SPR ($K_D = 12.24 \pm 4.05$ nM) (Supplementary Fig. 1). Both hydrophobic and hydrophilic interactions were found involving the residues in HCDR2, HCDR3, HFR3, LCDR1, and LCDR3 of P2C-1A3 and those in RBD (Fig. 2d). P2B-2F6 buried the smallest RBD interface (626 Å²) and showed the weakest binding affinity ($K_D = 15.62 \pm 2.94$ nM) among the three antibodies (Supplementary Fig. 1). As reported previously[14], the paratope consists of 11 heavy chain residues and 3 light chain residues. Among the three nAbs, P2C-1F11 buried the largest RBD areas, showed the strongest RBD binding affinity, and as noted, demonstrated the closest mimicry of the binding mode to ACE2. These attributes may jointly account for the most powerful neutralizing activity of P2C-1F11.

### The highest number of binding residues shared by P2C-1F11 and ACE2.
We conducted single-site alanine scanning mutagenesis for the 16 residues within the P2C-1F11 epitope to identify the key residues that mediate RBD binding. All mutated spikes were successfully expressed on the surface of HEK 293T cells, as measured by median fluorescence intensity (MFI) of the control S2 antibody through flow cytometry (Supplementary Fig. 2). Among the mutated residues, eight (T415A, Y421A,

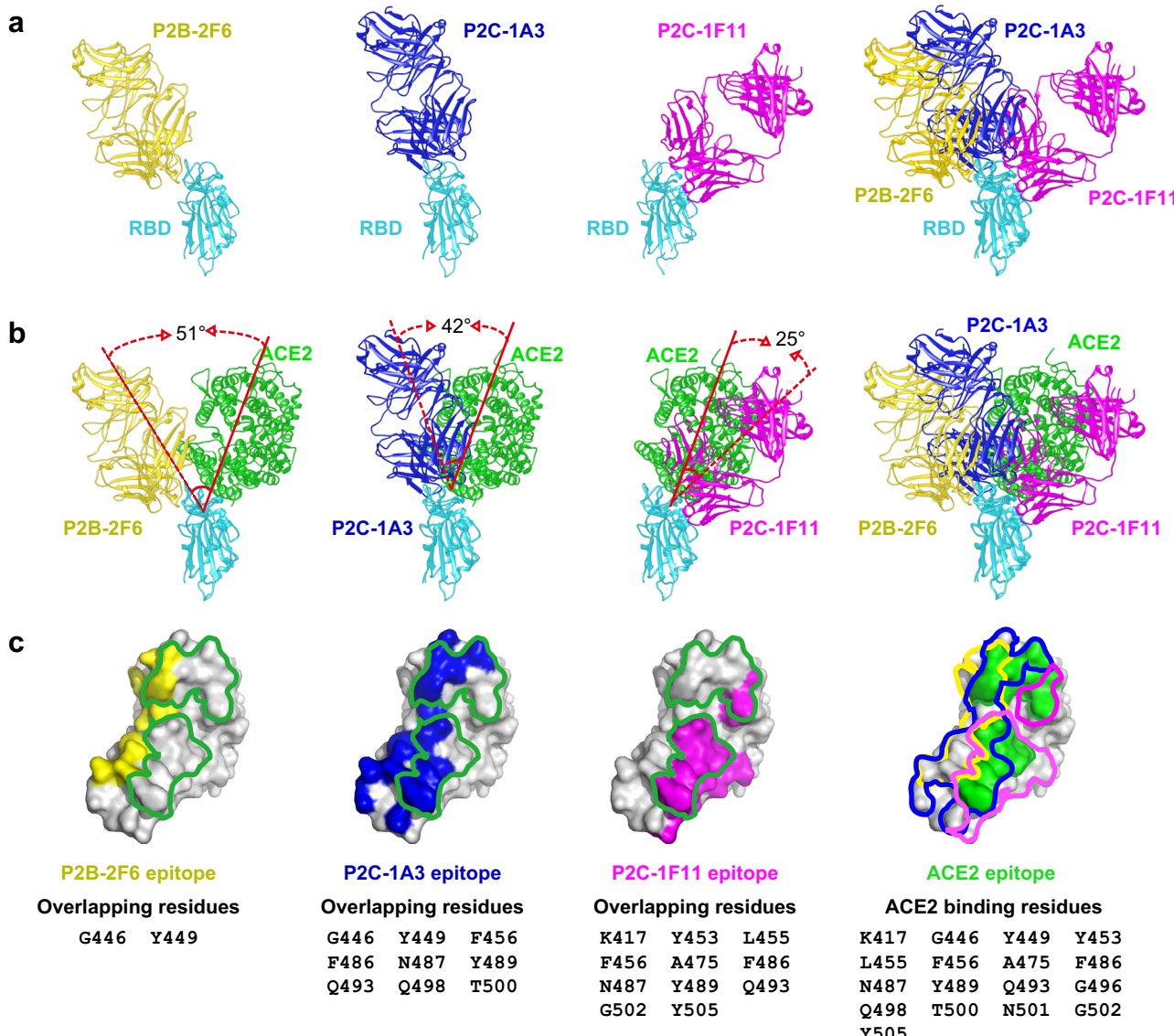

**Fig. 1 P2C-1F11 resembles ACE2 in binding to SARS-CoV-2 RBD. a** Overall structures of the SARS-CoV-2 RBD and Fab complexes. SARS-CoV-2 RBD is in cyan, P2B-2F6 Fab in yellow, P2C-1A3 Fab in blue, and P2C-1F11 Fab in magenta. The three Fab and RBD complexes are superimposed to demonstrate their relative positions and orientations. **b** Fab binding to RBD relative to RBD-ACE2 binding. The dashed and solid red lines indicate the long axes of Fabs and ACE2, respectively, in binding to RBD. Angles between the two are indicated. The three Fab and SARS-CoV-2 RBD complexes are superimposed to demonstrate their positions and orientations relative to ACE2. **c** The footprint of Fabs and ACE2 on SARS-CoV-2 RBD. Yellow, blue, magenta and green represent the footprint of P2B-2F6 Fab, P2C-1A3 Fab, P2C-1F11 Fab, and ACE2, respectively. Binding residues shared between each of the Fab and ACE2 are listed in the lower panel.

L455A, F456A, R457A, Y473A, N487A, and Y489A) resulted in more than 50% reduction in binding of P2C-1F11 (Fig. 3b and Supplementary Fig. 2). However, only four (L455A, F456A, N487A, and Y489A) appeared to affect P2C-1F11 specifically and resulted in over 95% binding reduction (Fig. 3a–d). The remaining mutations, however, resulted in a generalized effect of lower RBD binding with P2C-1A3, P2B-2F6, and ACE2 (Fig. 3a–d and Supplementary Fig. 2) and pseudovirus carrying these mutations demonstrated about 100-fold reduced infectivity (Fig. 3e). Interestingly, P2C-1F11 and ACE2 shared similar binding patterns, particularly for four mutants (L455A, F456A, N487A, and Y489A) (Fig. 3a, b). But some differences do exist for other residues, such as F486A, Q493A, and Y505A. The most dramatic was the Y505A mutation, which resulted in the complete loss in ACE2 binding while RBD binding with P2C-1F11 remained virtually unaffected. In contrast, Q493A enhanced

ACE2 binding about fourfold but only slightly improved for P2C-1F11 binding (Fig. 3a, b). Furthermore, the critical role of these four residues on P2C-1F11 binding is further supported by the SPR (Supplementary Fig. 3). The recombinant RBDs bearing these single mutations resulted in a significant reduction in P2C-1F11 binding, ranging from about 20-fold to complete loss compared to the wild-type control (Supplementary Fig. 3). Interestingly, all four residues are recognized by the heavy chain of P2C-1F11. The amino acids at L455, F456, and Y489 have hydrophobic interactions with P2C-1F11, while Y489 and N487 contribute multiple hydrogen-bond interactions for P2C-1F11 recognition (Fig. 2a, b).

For P2C-1A3, only two (F456A and Y489A) of the four mutations (L455A, F456A, N487A, and Y489A) substantially reduced RBD binding (Fig. 3c). Additional F486A mutant is particularly detrimental to P2C-1A3 and almost eliminated

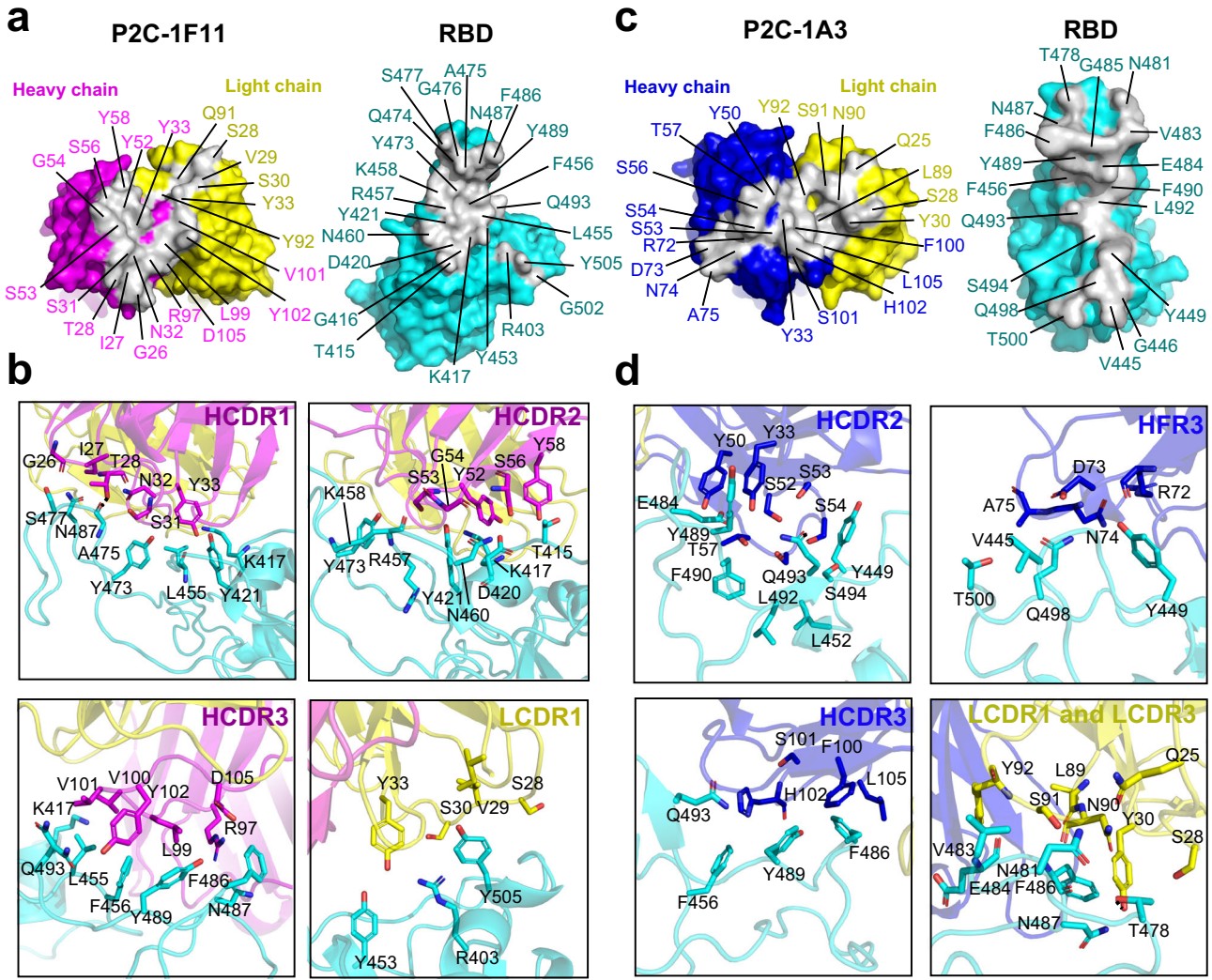

**Fig. 2 The binding interface between P2C-1F11 Fab, P2C-1A3 Fab, and SARS-CoV-2 RBD. a** The paratope and epitope of P2C-1F11. The paratope is in gray and binding residues within are indicated in either magenta or yellow, depending on their origin from heavy or light chain Fab. The SARS-CoV-2 RBD epitope is also gray and antibody binding residues within are highlighted in cyan. **b** Interactions between the P2C-1F11 Fab and SARS-CoV-2 RBD. **c** The paratope and epitope of P2C-1A3. The paratope is in gray and binding residues within are indicated either in blue or in yellow, depending on their origin from heavy or light chain Fab. The SARS-CoV-2 RBD epitope is also in gray and binding residues within are highlighted in cyan. **d** Interactions between P2C-1A3 Fab and SARS-CoV-2 RBD.

binding, consistent with the epitope residues defined by structural analysis (Figs. 2d, 3c, and Supplementary Fig. 2). However, P2B-2F6 was the least impacted and none of these four residues mutations had any discernable effect on its binding activity (Fig. 3d). Collectively, these results highlight that P2C-1F11 shares a greater number of key binding residues with ACE2 than P2C-1A3 or P2B-2F6. Lastly, a comparison of residues in the nAb epitopes for SARS-CoV-2 and SARS-CoV reveals substantial degrees of diversity. Only 10 of 23 residues for P2C-1F11, 5 of 18 of P2C-1A3, and 4 of 13 of P2B-2F6 are conserved, providing an atomic explanation for the lack of cross-reactivity to SARS-CoV RBD (Supplementary Fig. 4).

We also studied the impact of these mutations on antibody neutralization. Pseudoviruses bearing mutated S proteins were subjected to serial dilutions of each nAb to evaluate changes in neutralization sensitivity. Consistent with binding analysis, the four mutated residues (L455A, F456A, N487A, and Y489A) resulted in complete resistance to P2C-1F11 neutralization (Fig. 3f). But only two (F456A and Y489A) rendered complete resistance to P2C-1A3 neutralization (Fig. 3g). F486A appeared to

confer complete resistance specifically to P2C-1A3 but had no significant effect on P2B-2F6 (Fig. 3f–h). None of the four mutants, however, changed the neutralization profile of P2B-2F6 (Fig. 3h). Of note, four mutations (D420A, N460A, F486A, and Y505A) that affected P2C-1F11 neutralization had no impact on binding to cell surface-expressed S (Fig. 3f, b), suggesting that determinants of neutralization are more complex than binding alone can account for.

**Shedding of S1 triggered by P2C-1F11.** It is known that antibody binding can trigger dissociation of the S1 subunit and "shedding" from the membrane-bound S glycoprotein, rendering it non-functional for viral entry[5,18,30]. To investigate whether and to what extent the three nAbs trigger S1 shedding, we incubated cell-surface expressed S glycoprotein with a saturated concentration of testing antibodies and measured their binding over time by flow cytometry. As shown in Fig. 4a and Supplementary Fig. 5, P2C-1F11 triggered substantial levels of S1 shedding as indicated by the rapid and substantial decline in MFI over the

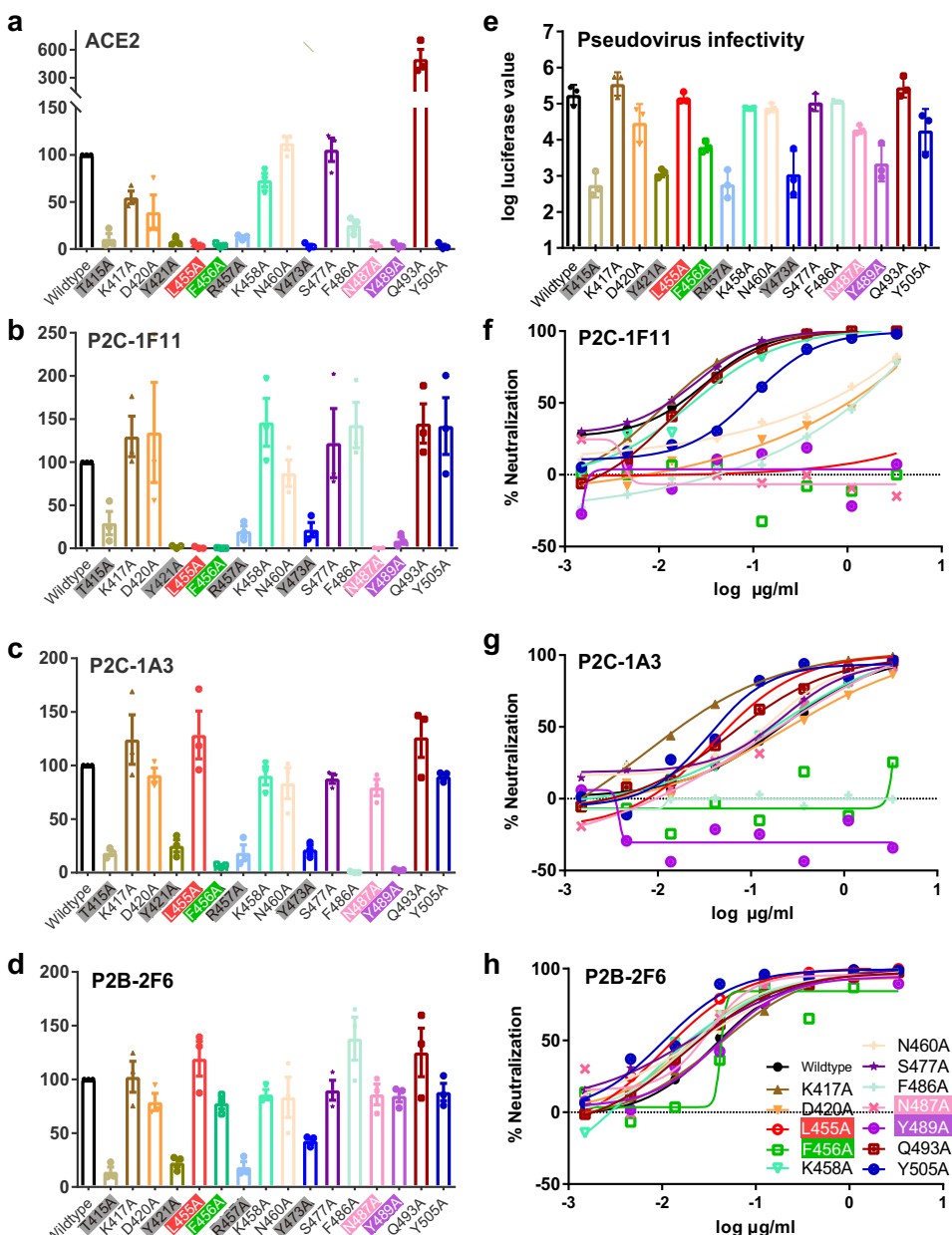

**Fig. 3 Impact of key residues on antibody binding and neutralization.** Percent changes in mean fluorescence intensity (MFI) to surface-expressed mutated SARS-CoV-2 S glycoprotein relative to those of wild-type analyzed by **a** recombinant ACE2, **b** P2C-1F11, **c** P2C-1A3, and **d** P2B-2F6. All MFI values were normalized relative to that of the S2-specific antibody. Impact of mutated residues on pseudovirus infectivity (**e**) and on neutralization sensitivity to **f** P2C-1F11, **g** P2C-1A3, and **h** P2B-2F6. Residues that adversely affected viral infectivity are in gray boxes while those that specifically yet differentially affected binding and neutralization are in red, green, pink, and purple boxes. Data shown in (**a–e**) are from three independent experiments. Data are presented as mean values ± SEM. Data shown in **f–h** are representative of two independent experiments.

120 min incubation period. On the other hand, P2C-1A3 and P2B-2F6 demonstrated little such capacity. Importantly, no concomitant decline was found in MFI of S2-specific antibody over the course of P2C-1F11 incubation (Fig. 4c and Supplementary Fig. 5), indicating that only S1 was shed off while S2 remained on the cell surface. Consequently, the MFI ratio between S1/S2 declined rapidly (Fig. 4e). On the other hand, the GSAS-containing spike was relatively stable without obvious changes either treated with S1 or S2 specific antibodies, suggesting fully cleaved S1 is required for its shedding off from the cell surface (Fig. 4b, d, f). Furthermore, this finding was consistent across the three experimental conditions of either 4, 37, or 37 °C with extra Furin protease expressed in the cells to produce more

cleaved S1 on the cell surface although the dynamics and degree of shedding triggered by P2C-1F11 varied among the three conditions (Supplementary Fig. 6). The greatest and most rapid shedding was found for cells with extra Furin expression at 37 °C, for which there was an 80% reduction in MFI at 60 min relative to that at 5 min time point (Supplementary Fig. 6c, g). Shedding of S1 triggered by P2C-1F11 on cells without extra Furin expression at 37 °C was also obvious but to a less extent (Supplementary Fig. 6a, e). Lowering the temperature from 37 to 4 °C attenuated the shedding process, suggesting antibody-triggered shedding is temperature-dependent (Supplementary Fig. 6b, f). Lastly, western blotting analysis of S1 in the cell supernatant demonstrated the rapid shedding occurred from 5 to 30 min after incubation

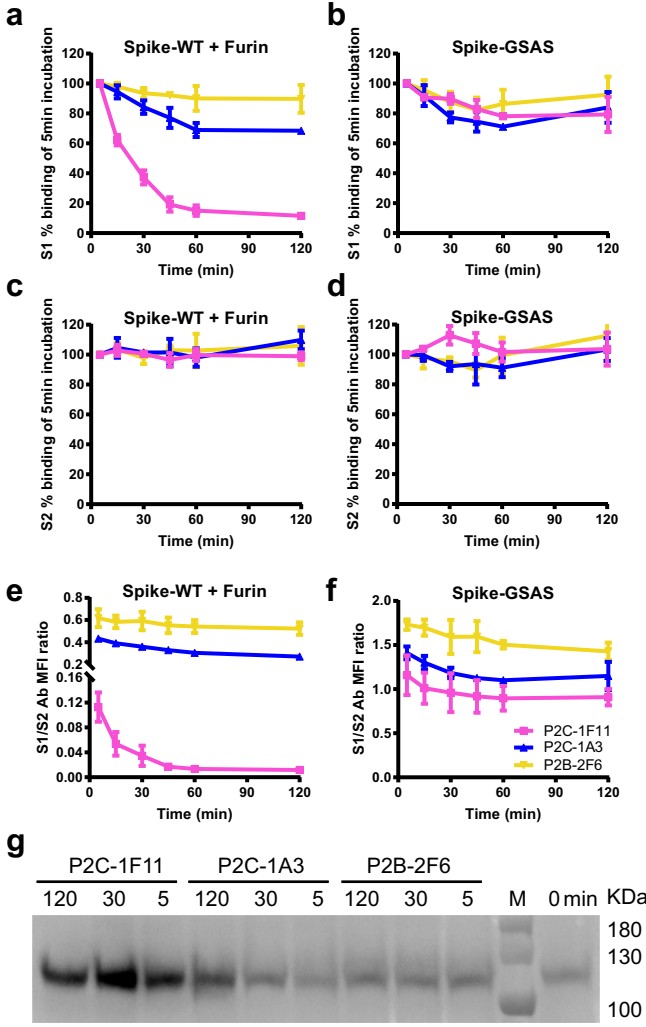

**Fig. 4 Shedding of S1 from cell surface-expressed S triggered by P2C-1F11. a–f** MFI dynamics of S1, S2, and S1/S2 ratios measured by binding signals of the testing antibodies. The percentage of positive cells at each allocated time point was determined by the MFI weighted by multiplying the number of positive cells and normalized in relative to that at 5 min time point. **g** Western blotting analysis of S1 shedding into the supernatant. Data shown were from three independent experiments. Data are presented as mean values ± SEM.

with P2C-1F11 followed by a relatively stable process up to 120 min (Fig. 4g). No such changes were found when cells were treated with P2C-1A3 or P2B-2F6. Only after treatment with P2C-1A3 for 120 min, a slight increase in S1 became detectable compared to the background control of spontaneous shedding of S1 in the absence of antibody (0 min). These results highlight the unique and exceptional ability of P2C-1F11 in triggering the shedding of S1 compared to P2C-1A3 and P2B-2F6.

**P2C-1F11 protects hACE2-sensitized mice from SARS-CoV-2 infection.** Next, we evaluated the protective potential of P2C-1F11 against SARS-CoV-2 infection in a well-established Ad5-hACE2-sensitized mice model[31]. A total of 36 BALB/c mice transduced with Ad5-hACE2 were equally divided into the prophylactic and therapeutic groups. For the prophylactic group, a set of nine mice were intraperitoneally administered with either P2C-1F11 or negative control antibody VRC01 at a dose of 20 mg kg$^{-1}$ body weight one day before the intranasal challenge with $1 \times 10^5$ PFU SARS-CoV-2 live virus. For the therapeutic group,

the regimen is the same except the antibodies were administered 2 h after viral challenge. Five of the animals were monitored for body weight throughout the 6-day observation period while the remaining four were sacrificed for lung tissues 3 days after viral challenge. As shown in Fig. 5, the prophylactic and therapeutic use of P2C-1F11 were able to reduce lung viral titer for about four-logs compared to that in the control animals. P2C-1F11 treated animals also maintained relatively stable body weight whereas those treated with the negative control antibody VRC01 experienced a dramatic loss of body weight starting from three days after viral infection. These results indicate that the potent neutralizing activity of P2C-1F11 was translated into protectivity in vivo for both prophylactic and therapeutic interventions.

## Discussion

We report here the structural and functional basis for the distinct neutralizing activities of three nAbs isolated from SARS-CoV-2 convalescent individuals. From a structural perspective, we provide and compare the atomic details of the epitopes, paratopes, the angles and degrees of potential clashes with receptor ACE2 in binding to RBD. The most potent nAb, P2C-1F11, shares the closest RBD-ACE2 binding mimicry, the greatest number of epitope residues, and most spatial clashing with receptor ACE2. Taken together, these features provide the structural basis for P2C-1F11 to effectively compete with ACE2 for binding to RBD. From the functional perspective, P2C-1F11 is more similar to ACE2 in its binding profile to the S glycoprotein compared to P2C-1A3 and P2B-2F6. In particular, P2C-1F11 is the only antibody studied here capable of triggering substantial dis-association of S1 from the cell surface S glycoprotein. It is reasonable to speculate that all these unique features combined contribute to the potent neutralization activity and strong in vivo efficacy of P2C-1F11. Of the nAbs investigated, P2C-1F11 is most likely to exert its antiviral activity through functional mimicry of receptor ACE2.

It has been speculated that the S glycoproteins of SARS-CoV-2 could be a moving target for antibody recognition. Fortunately, we have not seen a substantial degree of sequence and structural variability among the S glycoproteins submitted to the database. However, mutated residues conferring resistance have been generated through in vitro selection[19], highlighting the potential risk in the emergence of antibody-resistant strains. A major question remains on how and whether nAbs could respond to such mutations to provide immunity. One solution is to cover as much of the critical RBD-ACE2 interface as possible to maximize inhibitory potential. The distinct yet overlapping RBD epitopes among P2C-1F11, P2C-1A3, and P2B-2F6 support this hypothesis. Another solution is to target regions beyond the RBD-ACE2 interface, which may provide indirect means to disrupt the viral entry process. This theory is supported by the recent identification of nAbs against either the NTD or the quaternary epitope between NTD and RBD, and even the core domain of RBD[16,22,24]. A third solution would be to target as many vulnerable conformations as possible given the structural variability of the S glycoprotein during viral entry[13,19]. To this end, we compared different modes of recognition by docking the three antibodies onto the S trimer in the prefusion state. Interestingly, P2C-1F11, like ACE2, preferentially recognized RBD in the "up" conformation, whereas P2C-1A3 and P2B-2F6 bound to both "up" and "down" conformations (Supplementary Fig. 7). In any given infected individual, multiple nAbs targeting different conformations and epitopes are expected to work in concert to minimize or prevent viral escape from neutralization. This also implies that SARS-CoV-2 would have to acquire mutations in all major epitopes to fully escape antibody recognition in vivo. In

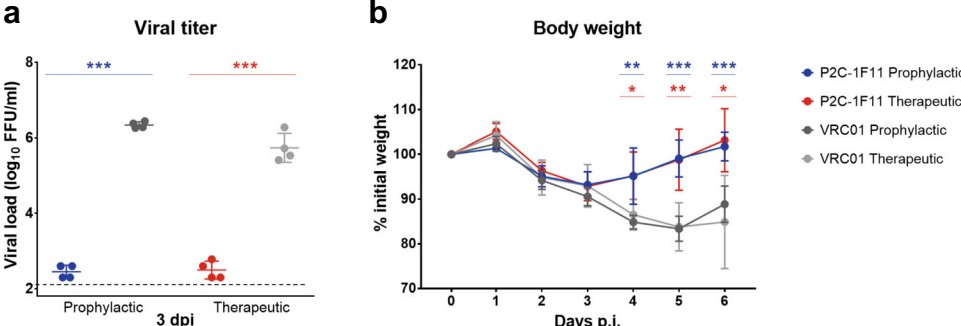

**Fig. 5 Protective potential of P2C-1F11 against SARS-CoV-2 infection in the Ad5-hACE2-sensitized mice.** Ad5-hACE2 transduced BALB/c mice were administered with P2C-1F11 either one day before (prophylactic group, n = 9) or 2 h after (therapeutic group, n = 9) challenge with live SARS-CoV-2. The viral load (n = 4) on day 3 post-infection (**a**) and the animal body weight (n = 5) throughout the 6-day time course (**b**) were monitored. The viral titer differences between the P2C-1F11 group and the VRC01 group are statistically significant with p value < 0.001. *p < 0.05; **p < 0.01; **p < 0.001. Data are presented as mean values ± SD.

this regard, combinations of two or more nAbs are expected to provide more efficacious immunity over a longer time. The structural and functional insights into the antibody recognition reported here will assist us in selecting and optimizing the best combination of antibodies for clinical interventions against SARS-CoV-2 infection.

## Methods

**Antibody and fab fragment production.** Antibody production was conducted as previously described[14]. Briefly, genes encoding the heavy and light chains of P2C-1F11, P2C-1A3, and P2B-2F6 were transiently transfected into HEK 293F cells using polyethylenimine (PEI) (Sigma). After 96 h, antibodies in the supernatant were collected and captured by Protein A-Sepharose (GE Healthcare). Bound antibodies were eluted and further purified by gel-filtration chromatography using a Superdex 200 High-Performance column (GE Healthcare). To produce Fab fragments, antibodies were cleaved using Protease Lys-C (Sigma) with an IgG to Lys-C ratio of 4000:1 (w/w) in 10 mM EDTA, 100 mM Tris-HCl, pH 8.5 at 37 °C for approximately 12 h. Fc fragments were removed using Protein A-Sepharose.

**Recombinant RBDs and receptor ACE2.** Recombinant RBDs and the N-terminal peptidase domain of human ACE2 (residues Ser19 to Asp615) were expressed using the Bac-to-Bac Baculovirus System (Invitrogen) as previously described[26]. Specifically, SARS-CoV-2 RBD (residues Arg319 to Lys529) containing the gp67 secretion signal peptide and a C-terminal hexahistidine was inserted into pFastBac-Dual vectors (Invitrogen) and transformed into DH10 Bac component cells (Supplementary Table 3). The recombinant bacmid was extracted and further transfected into Sf9 cells using Cellfectin II Reagents (Invitrogen). The recombinant viruses were harvested from the transfected supernatant and amplified to generate high-titer virus stock. Viruses were then used to infect Sf9 cells for RBD expression. Secreted RBD was harvested from the supernatant and purified by gel filtration chromatography.

**Crystallization and data collection.** Fab fragments were each mixed with SARS-CoV-2 RBD at a molar ratio of 1:1.2, incubated for 2 h at 4 °C, and further purified by gel-filtration chromatography. The purified complex was concentrated to 10 mg mL$^{-1}$ in HBS buffer (10 mM HEPES, pH 7.2, 150 mM NaCl) for crystallization. Screening trials were performed at 18 °C. The sitting drop vapor diffusion method was used by mixing 0.2 μL of protein with 0.2 μL of reservoir solution. Crystals of RBD–Fab complexes were successfully obtained in 0.1 M sodium citrate tribasic dihydrate, pH 5.8, and 20% PEG 6000. RBD-P2C-1A3 crystals were grown in 0.1 M lithium sulfate monohydrate, 0.1 M citric acid, pH 3.5, and 18% PEG 6000. Diffraction data were collected at the BL17U1 beamline of the Shanghai Synchrotron Research Facility (SSRF) and auto-processed with aquarium pipeline[32]. Data processing statistics are listed in Supplementary Table 1.

**Structural determination and refinement.** All structures were determined by the molecular replacement method using PHASER (CCP4 Program Suite)[33]. Search models were the SARS-CoV-2 RBD structure (PDB ID: 6M0J) and the heavy and light chain variable domain structures available in the PDB with the highest sequence identities. Subsequent model building and refinement were performed using COOT and PHENIX, respectively[34,35]. All structural figures were generated using PyMOL and Chimera[36,37].

**Angle and clash calculation between ACE2 and neutralizing antibodies.** The angle between neutralizing antibodies and SARS-CoV-2 RBD was calculated with

Chimera[37]. In detail, the axis of ACE2 or nAb was defined based on the mass center and the angle was calculated between the two long axes using the UCSF Chimera "define axis" and "angle" command. Once the RBD-ACE2 (PDB ID: 6M0J) was aligned to the RBD-nAb complexes, any residues that clashed were isolated and saved in a new coordinate file for volume calculation using the UCSF Chimera "measure volume" command.

**Surface plasmon resonance (SPR) experiments.** Antibodies were immobilized on a CM5 sensor chip (GE Healthcare) to ~500 response units (RUs) using a Biacore T200 (GE Healthcare) and a running buffer composed of 10 mM HEPES pH 7.2, 150 mM NaCl and 0.05% Tween 20. Serial dilutions of wild-type and mutated SARS-CoV-2 RBDs flowed through the sensorchip system. The resulting data were fitted to a 1:1 binding model using Biacore Evaluation Software (GE Healthcare).

**Antibodies binding to cell surface-expressed wild-type and mutated S glycoproteins.** HEK 293T cells were transfected with expression plasmids encoding either wild-type or mutated full-length SARS-CoV-2 S glycoproteins (Supplementary Table 3), and incubated at 37 °C for 36 h. Cells were removed from the plate with trypsin and distributed onto 96-well plates. Cells were washed twice with 200 μL staining buffer (PBS with 2% heated-inactivated fetal bovine serum (FBS)) between each of the following steps. First, cells were stained with 2 μg mL$^{-1}$ of nAb at 4 °C for 1 h in 100 μL staining buffer. Then, PE-labeled anti-human IgG Fc secondary antibody (Biolegend 410718) was added at a 1:40 dilution in 40 μL staining buffer at 4 °C for 1 h. After extensive washes, the cells were resuspended and analyzed with FACS Calibur (BD Biosciences, USA) and FlowJo 10 software (FlowJo, USA) (Supplementary Fig. 8). HEK 293T cells with mock transfection were stained as background control. Antibody binding percentages were calculated by the ratio between mutated over wild-type MFI normalized in relative to that of S2 specific antibody (MP Biomedicals, Singapore 08720401). All MFI values were weighted by multiplying the number of positive cells in the selected gates.

**Neutralization of wild-type and mutated SARS-Cov-2 pseudoviruses.** SARS-CoV-2 pseudoviruses were generated by co-transfecting HEK-293T cells (ATCC) with human immunodeficiency virus backbones expressing firefly luciferase (pNL4-3R-E-luciferase) and pcDNA3.1 (Invitrogen) expression vectors encoding either wild-type or mutated S proteins (Supplementary Table 3). Viral supernatant was collected 48 h later. Pseudoviruses were incubated with serial dilutions of nAbs at 37 °C for 1 h. Hela-hACE2 cells expressing human ACE2 protein were then added in duplicate to the mixture. Antibody neutralization percentages were determined by measuring luciferase activity in relative light units (Bright-Glo Luciferase Assay Vector System, Promega Bioscience) 48 h after exposure to virus-antibody mixture using GraphPad Prism 7 (GraphPad Software Inc.).

**Detection of S1 Shedding from cell surface-expressed SARS-CoV-2 S glycoprotein.** HEK 293T cells expressing wild-type or mutated SARS-CoV-2 S protein on the cell surface were generated as described above. MFI dynamics of S1 and S2 over the testing antibodies incubation were conducted under these two conditions: (1) at 37 °C with co-transfection of the plasmid encoding Furin protease to enhance the cleavage between S1 and S2; and (2) at 37 °C with a mutated S containing GSAS, substituting RRAR at the junction between S1 and S2 to avoid digestion by Furin protease. The cells were incubated with nAbs for 120, 60, 45, 30, 15, or 5 min, and an S2-specific antibody (MP 08720401). Immediately after the allocated incubation time, antibody-stained cells were transferred to ice and thoroughly washed with ice-cold PBS and 2% FBS. Samples were stained with the anti-human IgG PE (Biolegend 410718) and anti-mouse IgG APC secondary antibody (Biolegend 405308)

simultaneously. After a thorough wash with ice-cold PBS and 2% FBS, samples were resuspended and analyzed with FACS Calibur (BD Biosciences, USA) and FlowJo 10 software (FlowJo, USA) (Supplementary Fig. 8). The MFI of S1, S2, and S1/S2 ratio were recorded weighted by multiplying the number of positive cells in the selected gates and normalized in relative to that at the 5-min time point.

To confirm the finding and explore the factors that influence the shedding process, we studied the dynamics of S1 shedding under four distinct conditions: (1) at 37 °C; (2) at 4 °C; (3) at 37 °C with co-transfection of the plasmid encoding Furin protease; and (4) at 37 °C with a mutated S containing GSAS. Cells were stained with either ACE2, the testing antibodies, or control S2 antibody. Samples were prepared in multiples for serial incubations at 37 °C or 4 °C for 240, 120, 60, 30, 15, or 5 min. Samples were then stained with one of the following secondary antibodies: anti-his PE (Miltyni 130120787) for ACE2, anti-human IgG Fc PE (Biolegend 410718) for nAbs, or anti-mouse IgG Fc FTIC (ThermoFisher A10673) for S2 antibody (MP 08720401).

**Western blot analysis of S1 in the cell supernatant**. The S1 protein shed into the supernatant was detected by western blotting. In brief, approximately $10^5$ 293T cells transfected with spike expression vector were incubated with testing antibodies for 120, 30, 5, or 0 min. All samples were incubated at 37 °C for 120 min to ensure the cells were treated under the same condition except the testing antibody incubation. The S1 protein-containing supernatant was cleared for cell debris by centrifugation, run on a 4–12% gradient Tris/MOPS-Gel (GenScript), and transferred to polyvinylidene fluoride membranes. An anti-S1 SARS-CoV-2 S polyclonal antibody (1:2000 dilution; Sino Biological, Cat#40150-T62) and horse-radish peroxidase (HRP)-conjugated goat anti-rabbit secondary antibody were used for Western blotting.

**Antibody protection in hACE2 transduced mice**. All protocols for animal experiments were approved by the Institutional Animal Care and Use Committees of the Guangzhou Medical University. All work with live SARS-CoV-2 was conducted in the Biosafety Level 3 (BSL3) Laboratories. Mice were lightly anesthetized with isoflurane and transduced intranasally with $2.5 \times 10^8$ FFU of Ad5-hACE2 in 75 mL DMEM 5 days before antibody protection experiments.

For the prophylactic group, P2C-1F11 or negative control VRC01, an anti-HIV-1 antibody, was administered through intraperitoneal injection at a dose of 20 mg/kg one day before viral challenge. For the therapeutic group, the regimen was the same except the antibodies were given 2 h after viral challenge. Three days after the virus challenge, lung tissues were harvested to quantify the viral load, and body weight was monitored over a 6-day time course.

The virus in the lung homogenate was titrated by the focus forming assay. Briefly, the lung homogenate was serially diluted and inoculated onto Vero E6 cells at 37 °C for 1 h. The inoculum was then removed before adding the overlay media (MEM containing 1.6% carboxymethylcellulose). After 24 h, cells were fixed with 4% paraformaldehyde and permeabilized with 0.2% Triton X-100. Cells were then incubated with a rabbit anti-SARS-CoV-2 nucleocapsid protein polyclonal antibody (Cat#40143-T62, Sino Biological, Inc. Beijing), followed by an HRP-labeled goat anti-rabbit secondary antibody (Cat#111-035-144, Jackson ImmunoResearch Laboratories, Inc. West Grove, PA). The foci were visualized by TrueBlue Peroxidase Substrate. Multiple $t$ tests were used to determine significant differences among the groups, and a $p$ value of <0.05 was considered statistically significant.

**Reporting summary**. Further information on research design is available in the Nature Research Reporting Summary linked to this article.

## Data availability

Data generated or analyzed during this study are included in this published article (and its supplementary information files). Source data are provided with this paper. Any other raw data pertaining to this study are available from the corresponding author upon reasonable request. The coordinates and structure factors files for the SARS-CoV-2 RBD-P2C-1F11 and SARS-CoV-2 RBD-P2C-1A3 complex were deposited to Protein Data Bank with accession code 7CDI and 7CDJ, respectively. The sequences of P2C-1F11, P2C-1A3, and P2B-2F6 antibodies have been deposited in GenBank with accession code MW259136 for P2B-2F6 heavy chain, MW259137 for P2B-2F6 light chain, MW259138 for P2C-1F11 heavy chain, MW259139 for P2C-1F11 light chain, MW259140 for P2C-1A3 heavy chain, MW259141 for P2C-1A3 light chain.

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

## Acknowledgements

We thank the SSRF BL17U1 beamline for data collection and processing. We thank the X-ray crystallography platform of the Tsinghua University Technology Center for Protein Research for providing the facility support. We also thank Dr. Fei Gao for initial coordinating and assisting in the design of animal experiments. This work is supported by funds from the Beijing Advanced Innovation Center for Structural Biology and the National Key Plan for Scientific Research and Development of China (2020YFC0848800, 2020YFC0844200, 2018ZX10731101-002, 2017ZX10201101-001-003, and 2016YFD0500307), the Science and Technology Innovation Committee of Shenzhen Municipality (202002073000002). It is also supported by Tsinghua University Initiative Scientific Research Program (20201080053), the National Natural Science Foundation Award (81530065 and 91442127), Beijing Municipal Science and Technology Commission (171100000517-001 and -003), Tsinghua University Spring Breeze Fund (2020Z99CFG004), and Tencent Foundation, Shuidi Foundation, TH Capital and the National Science Fund for Distinguished Young Scholars (82025022).

## Author contributions

J.G. and R.W. carried out protein expression, purification, crystallization, diffraction data collection, structural determination, cell staining, and neutralization with the help of Q.Z. and B.J., P.C., Y.T., S.S., L.C., B.Z., S.S., J.J.Z., H.W., and X.S. helped in antibody characterization, protein expression, purification, cell staining, and pseudovirus neutralization. J.S. and J.C.Z. designed and performed animal experiments. S.Z. helped in crystallization and diffraction data collection. Q.D. and L.L. provided critical support for reagents and guidance to the project. Z.Z., X.W., and L.Z. conceived, designed, directed the study, and wrote the paper.

## Competing interests

The authors declare no competing interests.
