## [Peer Review File · Nature Communications]

REVIEWER COMMENTS

Reviewer #1 (Remarks to the Author):

The manuscript by Ge et al. describes a very important study on understanding of the structure and the mechanism of action of neutralizing antibodies isolated from SARS-CoV-2 infected individuals. Such studies are critical to provide structure and function information to the efforts focused on designing therapeutic antibodies. The paper is very well written, it's concise and to-the-point. It is worth of publication in Nat Commun with some minor revisions.

Two neutralizing antibodies were crystallized with SARS-CoV-2 RBDs and the structures of the complexes determined at 3 and 3.4Å resolutions, which are good enough resolutions to discuss the overall complex structure. The resolutions, however, may not be good enough for detailed discussion of hydrogen bonding interactions, unless the residues involved are ordered and have strong electron density on their side chains. The authors, therefore, are encouraged to make a specific comment on this in the Results, especially when referring to Figs 2b and 2d. Also, if errors for SPR measurements are available they need to be added, because the values for P2C-1F11 and P2C-1A3 of 2.12 nM and 2.47 nM appear to be identical – it is generally known that SPR produces rather large errors in measurements and require multiple measurements to obtain a trustworthy KDs.

In the first paragraph of the Results, it would be necessary to provide an explanation for the “clash volume”. A general reader, and even a crystallographer with no experience in the structures of antibodies, might be confused. A couple of sentences to describe what the authors refer to when talking about clashes here would be most beneficial.

Reviewer #2 (Remarks to the Author):

Ge et al reported structural data of Fab-RBD complex from three neutralizing antibodies P2C-1F11, P2C-1A3 and P2B-2F6. The data for P2B-2F6 and RBD complex was reported previously (ref 14). Three antibodies were isolated from SARS-CoV-2 patients and their antigenic specificities and neutralizing functions were reported previously (ref 14). In the present paper, the authors reported the details of contact residues on the RBD by P2C-1F11 and P2C-1A3 and compared their interactive positions and orientations relative to ACE2. Antibody P2C-1F11 most closely mimicked binding of receptor ACE2 with RBD and displayed the most potent neutralizing activity. The key contact residues that are relevant to P2C-1F11 binding to RBD were studied by single-site alanine scanning mutagenesis, showing four RBD residues (L455, F456, N487 and Y489) are critical for the binding and

neutralization of antibody. The related analysis was also extended to the study of P2C-1A3 and P2B-2F6.

I think this study contributes to the human monoclonal antibody field for SARS-CoV-2 by a descent study design and structural analysis, although the proposed neutralizing mechanism is not convincing, the functional data with wild-type virus and in vivo data for P2C-1F11 are lacking.

A few minor comments are as follows,

1. The authors performed an experiment by incubating cell-surface expressed S glycoprotein with saturated concentration of antibodies and measured their binding over time by flow cytometry. What is the rationale that the rapid and substantial decline in binding signals along with the incubation time leads to the conclusion of S1 shedding?

2. Following the above question, i was wondering if the S1:S2 ratio is determined for the S domain following the incubation with different anti-RBD antibodies and in the control group?

3. Even though either the S1 shedding or unstable S protein is one of possible reasons for loss of binding signals, why the effect greatly varied among three anti-RBD (ACE2 blockage) antibodies?

Reviewer #1

The manuscript by Ge et al. describes a very important study on understanding of the structure and the mechanism of action of neutralizing antibodies isolated from SARS-CoV-2 infected individuals. Such studies are critical to provide structure and function information to the efforts focused on designing therapeutic antibodies. The paper is very well written, it's concise and to-the-point. It is worth of publication in Nat Commun with some minor revisions.

1. Two neutralizing antibodies were crystallized with SARS-CoV-2 RBDs and the structures of the complexes determined at 3 and 3.4Å resolutions, which are good enough resolutions to discuss the overall complex structure. The resolutions, however, may not be good enough for detailed discussion of hydrogen bonding interactions, unless the residues involved are ordered and have strong electron density on their side chains. The authors, therefore, are encouraged to make a specific comment on this in the Results, especially when referring to Figs 2b and 2d.

Response: We appreciate the reviewer's comments and encouragement. We agree with the reviewer that 3-3.5 angstrom is a marginal resolution for detailed discussion of hydrogen bonds unless the densities of the side chains are clear enough. Therefore, we reexamined the density of the side chains that form the hydrogen bonds between antibody and RBD indicated by PISA. As shown in Fig. S1 and S2 below, the density at the P2C-1F11/RBD interface is determined at 2.95 angstrom resolution, but that at the P2C-1A3/RBD interface is determined at 3.4 angstrom resolution. Therefore, to be more precise and consistent throughout, we decided not to describe nor discuss the hydrogen bonds for any of the antibodies studied here. The relevant text and Figure 2b and 2d in the revised manuscript have been edited to reflect such changes.

Fig. S1 Density map of residues forming the hydrogen-bond interactions at the interface of SARS-CoV-2 RBD/P2C-1F11. $2F_o - F_c$ electron density maps contoured at 1.0σ at the binding interface between the SARS-CoV-2 RBD (green) and P2C-1F11 (yellow for heavy chain and magenta for light chain) are shown.

Fig. S2 Density map of residues forming the hydrogen-bond interactions at the interface of SARS-CoV-2 RBD/P2C-1A3. $2F_o - F_c$ electron density maps contoured at 1.0σ at the binding interface between the SARS-CoV-2 RBD (green) and antibody P2C-1A3 (yellow for heavy chain and magenta for light chain) are shown.

2. Also, if errors for SPR measurements are available they need to be added, because the values for P2C-1F11 and P2C-1A3 of 2.12 nM and 2.47 nM appear to be identical – it is generally known that SPR produces rather large errors in measurements and require multiple measurements to obtain a trustworthy K_D s.

Response: The K_D values indicated in the original manuscript were derived from our previous study (Bin Ju., et al, 2020, Nature) where SARS-CoV-2 RBD was immobilized on the CM5 chip and over thirty antibodies were then flowed through in the SPR experiment. Following reviewers' suggestion and to mitigate the interference of bi-valent binding from both arms of antibody, we reversed the experimental procedure by immobilizing the three antibodies (P2B-2F6, P2C-1A3 and P2C-1F11) on the CM5 chip first and then flowed through the SARS-CoV-2 RBD. The same experiments were repeated three times. The estimated K_D values for P2B-2F6, P2C-1A3 and P2C-1F11 are 15.62 ± 2.94 nM, 12.24 ± 4.05 nM and 1.72 ± 0.84 nM, respectively (Fig. S3). These new K_D estimates have been updated in the revised manuscript and the raw SPR data were added as a supplementary figure (page 5, line 119-131).

B

	k_a (1/Ms)	k_d (1/s)	K_D (nM)	K_D (nM)
P2B-2F6-1	6.478E+6	0.09106	14.06	15.62 ± 2.94
P2B-2F6-2	7.407E+6	0.1022	13.80	
P2B-2F6-3	5.383E+6	0.1023	19.01	
P2C-1A3-1	1.115E+7	0.1029	9.23	12.24 ± 4.05
P2C-1A3-2	1.181E+7	0.1258	10.65	
P2C-1A3-3	6.985E+6	0.1177	16.85	
P2C-1F11-1	1.944E+6	0.002129	1.10	1.72 ± 0.84
P2C-1F11-2	1.905E+6	0.002625	1.38	
P2C-1F11-3	1.133E+6	0.003031	2.67	

Fig. S3 Measurement of K_D by SPR. (A) Binding curves of immobilized neutralizing antibody (P2B-2F6, P2C-1A3 and P2C-1F11) to the SARS-CoV-2 RBD. Data are presented by colored lines and the best fit of the data to a 1:1 binding model are shown in black. (B) Summary of the binding kinetics between SARS-CoV-2 RBD and the neutralizing antibodies measured in three independent experiments.

3. In the first paragraph of the Results, it would be necessary to provide an explanation for the “clash volume”. A general reader, and even a crystallographer with no experience in the structures of antibodies, might be confused. A couple of sentences to describe what the authors refer to when talking about clashes here would be most beneficial.

Response: According to the reviewer’s advice, we added a short explanation of clash volume (the total volume of clashed residues between P2C-1F11 and ACE2 when both would bind to the same RBD molecule) in the revised manuscript (Page 4, line 93-96). The calculation of clash volume was also explained in the Methods section “Angle and clash calculation between ACE2 and neutralizing antibodies” (Page 20, line 496-502).

Reviewer #2

Ge et al reported structural data of Fab-RBD complex from three neutralizing antibodies P2C-1F11, P2C-1A3 and P2B-2F6. The data for P2B-2F6 and RBD complex was reported previously (ref 14). Three antibodies were isolated from SARS-CoV-2 patients and their antigenic specificities and neutralizing functions were reported previously (ref 14). In the present paper, the authors reported the details of contact residues on the RBD by P2C-1F11 and P2C-1A3 and compared their interactive positions and orientations relative to ACE2. Antibody P2C-1F11 most closely mimicked binding of receptor ACE2 with RBD and displayed the most potent neutralizing activity. The key contact residues that are relevant to P2C-1F11 binding to RBD were studied by single-site alanine scanning mutagenesis, showing four RBD residues (L455, F456, N487 and Y489) are critical for the binding and neutralization of antibody. The related analysis was also extended to the study of P2C-1A3 and P2B-2F6.

I think this study contributes to the human monoclonal antibody field for SARS-CoV-2 by a descent study design and structural analysis, although the proposed neutralizing mechanism is not convincing, the functional data with wild-type virus and in vivo data for P2C-1F11 are lacking.

Response: We appreciate the reviewer’s comments and encouragement. To address the question raised by the reviewer, we conducted additional animal experiment to evaluate the protective potential of P2C-1F11 against SARS-CoV-2 infection in a well-established Ad5-hACE2-sensitized mice model. The results show that prophylactic and therapeutic use of P2C-1F11 were able to reduce viral titer and maintain relative stable body weight compared to the negative control antibody VRC01, suggesting the potent neutralizing activity in vitro could be translated into the protectivity in vivo in this particular animal model (Fig. S4). We have added the new protection data in the revised manuscript (Page 8, line 224 to page 9, line 241 and Page 18).

Fig. S4 Protective capacity of P2C-1F11 against SARS-CoV-2 infection in the Ad5-hACE2-sensitized mice model.

However, we agree with the reviewer that we are still uncertain about the neutralizing mechanism of P2C-1F11 in particular in the context of live virus. It may be due to the direct blocking of viral binding to receptor ACE2 and/or triggering premature shedding of S1, or some other unknown mechanisms yet to be discovered. We therefore performed additional experiments illustrated below to further confirm that triggering premature shedding of S1 is biochemically relevant, which may contribute to the overall potent and protective activity of P2C-1F11 (please see below).

A few minor comments are as follows,

1. The authors performed an experiment by incubating cell-surface expressed S glycoprotein with saturated concentration of antibodies and measured their binding over time by flow cytometry. What is the rationale that the rapid and substantial decline in binding signals along with the incubation time leads to the conclusion of S1 shedding?

Response: We were puzzled by the same question at the beginning and tried a couple of approaches to confirm the decline in binding signals was indeed a result of S1 shedding. One experiment was to analyze the dynamic changes of both S1 and S2 during incubation to see whether decline in fluorescent intensity is applicable to both S1 and S2 or simply just S1. If it was the former, decline in signal would be unlikely due to S1 shedding but to some other processes involving the entirety of spike such as antibody-triggered endocytosis of S1 and S2. However, if it was the latter, then antibody-triggered S1 shedding would be the most logical explanation. What the experiment results showed in the end was in complete agreement with the latter scenario (Fig. S5). While dramatic decline in fluorescent intensity was found for P2C-1F11 treated cells, that for S2-specific antibody remained relatively stable. Furthermore, such decline was almost abrogated when the cleavage site between S1 and S2 was inactivated by replacing with GSAS residues,

suggesting the presence of fully cleaved S1 was required for P2C-1F11 induced decline in fluorescent intensity. On the other hand, no such dramatic effect was found when incubated with P2C-1A3 and P2B-2F6 and only small levels of S1 reduction was found for both wildtype and GSAS spike although the underlying mechanism remains unclear (Fig. S5).

Fig. S5 Dynamic changes of S1 and S2 on the cell surface measured by S1- and S2-specific antibodies.

The other experiment was to monitor the dynamic changes of S1 in the cell supernatant throughout the incubation period. If S1 was indeed shed from the cell surface, it should become detectable and increase as the incubation time extended. In fact, this is exactly what we found. Increasing amount of S1 was detected in the cell supernatant from 5 to 30 min after incubation with P2C-1F11 and remained highly detectable up until 120 min (Fig. S6). In contrast, no such changes were found when cells were treated with P2C-1A3 or P2B-2F6. Only after treatment with P2C-1A3 for 120 min, a slightly increase in S1 became detectable compared to the background control. These results indicate that the decline in binding signals was indeed reflective of S1 shedding. P2C-1F11 is the most potent in triggering such event among the three antibodies studied here. In addition, spontaneous shedding of S1 also occurs in the absence of antibody, a rather unique feature compared to many other viral envelope glycoproteins.

Fig. S6 Antibody triggered S1 shedding to the cell supernatant detected by Western blot.

2. Following the above question, i was wondering if the S1:S2 ratio is determined for the S domain following the incubation with different anti-RBD antibodies and in the control group?

Response: As suggested by the reviewer, we measured the mean florescence intensity (MFI) of S1 and S2, and calculated S1/S2 ratio over incubation period. As shown in Fig. S7, P2C-1F11 treatment induced rapid decline in S1/S2 ratio of wildtype spike whereas no such effect was found for P2C-1A3 and P2B-2F6 over the course of incubation. For GSAS spike, however, three antibodies had fairly similar effect. These results support the notion that P2C-1F11 is more capable than P2C-1A3 and P2B-2F6 in triggering S1 shedding while S2 remain relatively unaffected. The original FACS data from where the MFI were derived is also shown (Fig. S8).

Fig. S7 Changes in S1/S2 ratio over incubation period measured by FACS.

Fig. S8 Shedding of S1 over time measured by flow cytometry.

3. Even though either the S1 shedding or unstable S protein is one of possible reasons for loss of binding signals, why the effect greatly varied among three anti-RBD (ACE2 blockage) antibodies?

Response: This is a great question and has been challenging our thinking as well. Although the three antibodies are all capable of competing with ACE2 for binding to RBD, P2C-1F11 is the closest mimicry of ACE2 in terms of angle of approach, binding footprint, and number of shared binding residues on the RBD surface. P2C-1F11 also demonstrated the highest binding affinity to RBD among the three antibodies. These attributes combined may provide some explanations for the extraordinary ability of P2C-1F11 in triggering S1 shedding and in neutralizing SARS-CoV-2. In other words, P2C-1F11 can exert its neutralizing activity not just through disrupting the proper interaction between RBD and ACE2 but also through inducing premature shedding of S1, thereby rendering the viral particles less infectious or completely loss its infectivity before even reaching to their target cells. Perhaps, this is the farthest we could extrapolate from our data and more work are certainly needed before we could fully understand the exact process and associated mechanism.

REVIEWERS' COMMENTS

Reviewer #1 (Remarks to the Author):

The authors have addressed my comments.

I commend the authors for re-doing SPR measurements in a better way and for additional animal studies.

Reviewer #2 (Remarks to the Author):

The authors have responded the requests and improved the manuscript.